# Mindfulness-Based Intervention for Caregivers of Frail Older Chinese Adults: A Study Protocol

**DOI:** 10.3390/ijerph19095447

**Published:** 2022-04-29

**Authors:** Herman H. M. Lo, Alma Au, W. V. Cho, Elsa N. S. Lau, Janet Y. H. Wong, Samuel Y. S. Wong, Jerf W. K. Yeung

**Affiliations:** 1Department of Applied Social Sciences, Hong Kong Polytechnic University, Hong Kong; hkaa2788@gmail.com; 2Caritas Medical Centre, Hospital Authority, Hong Kong; chowc@ha.org.hk; 3Department of Social Sciences, Education University of Hong Kong, Hong Kong; enslau@eduhk.hk; 4School of Nursing, University of Hong Kong, Hong Kong; janetyh@hku.hk; 5Jockey Club School of Public Health and Primary Care, Chinese University of Hong Kong, Hong Kong; yeungshanwong@cuhk.edu.hk; 6Department of Social and Behavioural Sciences, City University of Hong Kong, Hong Kong; jerf.yeung@cityu.edu.hk

**Keywords:** family caregivers, frail older adults, mindfulness-based intervention, randomized controlled trial, Chinese intergenerational caregiving

## Abstract

Studies have consistently showed that informal caregivers have worse health, more medical consultations, anxiety and depression, and lower quality of life than those who do not provide such care. Positive outcomes of psychoeducation interventions have been found, but many of them are relatively long in duration, making them less cost-effective in implementation. The proposed study is a multi-site, three-arm randomized controlled trial of a mindfulness-based intervention for Chinese family caregivers. Effects of the intervention will be compared with those of an evidence-based psychoeducation program and treatment-as-usual. Two hundred forty cross-generational caregivers of frail older adults with moderate to severe levels of frailty will be recruited and randomly assigned to mindfulness-based intervention, psychoeducation, and treatment-as-usual experimental conditions. Program effectiveness will be analyzed on measures of caregiver burden, depression, anxiety, positive caregiving experience, spirituality, family conflict, and the biomarker of heart rate variability. Measures on coping styles, experiential avoidance, and self-efficacy will be explored to see if they mediate the changes to participant improvements in outcomes. Six-month follow-up will be included to investigate the maintenance effects. This study will provide evidence on mindfulness-based interventions on caregivers of frail older adults and expand the existing models of intergenerational caregiving in Chinese culture.

## 1. Introduction

### 1.1. Burden of Family Caregivers of Frail Elders in an Ageing Society

Frailty refers to a multi-dimensional syndrome that includes the loss of physical abilities and the accumulation of deficits due to ageing. Based on an estimation in Canada, it increases steadily with age and its prevalence rises from 22% for people aged between 65 to 69 to 44% for people aged 85 years and above [1]. The gradual and cumulative decline in several physiological systems and the associated vulnerabilities are associated with higher risks of disability, falls, hospitalizations, and death [2]. Frail persons were 8 times more likely to have dementia than healthy counterparts [3].

In Hong Kong, 1.95 million persons among the total population are living with disabilities or chronic diseases, and the number of family caregivers has shown a drastic increase of 39% to 229,000 from 2006 to 2013 [4]. About 30% are adult children or extended family members, serving as primary informal caregivers without domestic helpers. Along with the demands of caregiving for a frail family member, the adult children of the elderly experience family and role conflicts [5]. Primary informal caregivers have worse health profile, more medical consultations, anxiety and depression, weight loss, and lower quality of life compared with their counterparts [6]. Similarly, Western studies have consistently reported that perceived burden and depressive symptoms of caregivers are associated with severe impairments of care recipients, and adult children living together as caregivers experiencing role conflicts reported highest level of burden [7]. It is also alarming that abusive behaviors towards the elderly with dementia were predicted by anxiety and depression of caregivers, as such association was mediated by extended hours of care and caregiving burden [8].

There has been concern about caregiver coping strategies, but mixed results were found in the outcomes of traditional classification of problem-focused and emotion-focused coping [9,10]. Problem-focused coping refers to strategies that target solutions for the problem that create distress, while emotion-focused coping involves the management of emotional distress caused by stressors. It should be noticed that emotion-focused coping is very diverse and heterogeneous, including wishful thinking, avoidance, self-blame, spirituality, and seeking social support. For caregivers of family members with dementia, problem-focused coping was associated with less burden, and avoidance coping predicted greater distress [11]. Although caregivers frequently adopted problem-focused strategies in general, female caregivers tended to use more emotion-focused strategies that were associated with higher levels of distress [12]. However, some studies have suggested that problem-focused strategies may be less helpful to improve the elderly’s situation, and emotion-focused strategies, such as acceptance, getting emotional support, and positive reframing were found to predict lower severity in depression and anxiety of caregivers [13]. In a longitudinal study of caregiver coping strategies, an association of wishful thinking and self-blame with caregiver anxiety and male caregiver health problems was found [14]. These studies suggest coping strategies should be targeted in caregiver intervention.

### 1.2. Intervention for Managing Caregiver Burden of Frail Older Adults

Caregiver burden refers to the perceived adverse effects of caregiving on the emotional, social, financial, physical, and spiritual functioning of the caregivers [15]. To improve the lives of community-dwelling frail older adults and their informal caregivers, effective support is required to reduce the demand for institutional care and hospital admissions. Psychological interventions have been useful in strengthening family caregiving and effective intervention components, which include information about problem behaviors and diseases faced by the elderly, changing caregivers’ perceptions about such problems, improving problem-solving skills, stress reduction strategies, improving family support, and exploring care plans with formal help [16]. Well-designed and clearly structured interventions improved the psychological health of caregivers including depression, quality of life, attitudes towards caregiving, and anxiety [17]. Caregiver coping interventions were particularly effective in improving their mental health, and they could be implemented either individually or in a group [18].

Mindfulness-based interventions (MBIs) have been widely adopted as an evidence-based approach in supporting people with chronic medical conditions [19]. Kabat-Zinn [20] defined mindfulness as the awareness that emerges through paying attention purposefully in the present moment and non-judgmentally. In MBI, instructors provide guided training to develop healthy coping skills and people can be more aware of some habitual and maladaptive use of problem-focused coping [20,21]. Specific exercises can help to increase participant’s awareness of their patterns of stress reactions that characterize emotional distress, and coping strategies including acceptance or accommodation can be enhanced. In MBI, caregivers can learn skills to manage their burden, by integrating mindfulness exercises with caregiving-specific psychoeducation. However, there have not been adequate studies investigating whether MBI creates changes to coping styles and whether it can explain the positive outcomes of MBI.

A few studies of MBIs for family caregivers reported small effect sizes of positive outcomes, but they had weak research designs, heterogeneous participants, small sample sizes, and/or the lack of control groups [22]. Many studies in this area have been based on a benchmark MBI, an 8-week Mindfulness-Based Stress Reduction (MBSR) program that lasts over 20 h [23,24]. In a large trial, 141 Chinese informal caregivers were randomized into an MBSR or a self-help control group [25]. MBSR participants had greater decreases in depressive symptoms at post-intervention and 3-month follow-up than those of the control group, and a larger improvement in self-efficacy was found in MBSR group at 3-month follow-up. In view of the difficulties in recruitment and attrition of caregivers as well as the pragmatic considerations regarding the implementation of health care practice, a low-intensity MBI will likely be more suitable for adult children caregivers who have been burdened by multiple family roles. An early pilot study modified the original MBSR to a 7-week, brief program that lasted for 10.5 h, and 31 caregivers were randomized into an MBI, a psychoeducation, or a respite care program [26]. Results indicated that both MBI and psychoeducation interventions reported decreases in distress and avoidance coping and increased self-efficacy. The major limitations of the above two studies were the heterogeneity of the caregivers who were involved in caregiving of family members with different diagnoses and consisted of spouse and adult children [25], and the small sample size that limited the power to detect the specific outcome efficacy and mediators of change for each intervention [26].

### 1.3. A Call for Model for Intergenerational Caregiving

The stress and coping model emphasize the role of one’s appraisals of their stressful events and an individual’s attempts to master, minimize, or tolerate the stress induced by a perceived challenge, harm, or threat [27]. In earlier years, it was common to categorize different coping styles as problem-focused vs. emotion-focused. However, there has been some criticism that such classification is overly simplistic that cannot capture the diversity of coping [28]. In a recent scale validation study, coping styles were represented in an 11-factor model: accommodation, self-blame, behavioral disengagement, and denial predicted higher levels of depression. Support seeking predicted lower level of depression. Self-blame, denial, and behavioral disengagement predicted higher levels of anxiety. Accommodation predicted lower level of anxiety, and problem-solving predicted neither depression nor anxiety [29].

A few studies have investigated mediators of the outcome of a caregiving intervention. Psychoeducation interventions were often based on the assumption that caregivers would benefit from structured knowledge about caregiving and should apply them into their caregiving problems based on positive values of problem-focused coping [30]. However, many intervention studies have either failed to investigate coping strategies of caregivers or assigned unclear labels to positive or negative coping styles but were too reluctant to define them clearly [31]. A meta-analysis of coping styles and mental health concluded that dysfunctional coping predicted high levels of depression and anxiety, acceptance-based coping and emotional support predicted low levels of depression and anxiety, and problem-focused coping was not associated with mental health outcomes [32]. In another studies of dementia caregiver intervention, a more complicated picture emerged, as different coping styles may be associated with positive outcome changes in different studies. For examples, caregivers participated a cognitive behavioral intervention and reported significantly more both rational problem-solving and emotion-focused coping than did their counterparts [33]. In another recent trial, emotion-focused coping mediated the improvements of depression and anxiety only among caregivers with higher pre-treatment distress, problem-focused coping remained unchanged, and dysfunctional coping did not decrease after intervention [34,35].

Although the concept of coping is fundamental to MBIs, the relationships between outcomes of MBI and coping are understudied. MBIs have also led to reductions of impulsive and reactive coping style of heart disease patients [36] and an increase in problem-focused coping and a reduction in avoidance-based coping in university student participants [37]. To better understand the coping styles of Chinese caregivers, the proposed study will investigate the coping models of MBI and caregiving psychoeducation, using a comprehensive, multi-dimensional construct of coping [29].

### 1.4. Outcomes and Mediators of Intergenerational Caregiving

In addition to caregiver mental health, as measured by depression, anxiety, and caregiver burden, other possible benefits will be explored in this study. There has been a growing interest in the physiological outcomes of a stress reduction program using a biomarker, such as heart rate variability (HRV), but their use in caregiver studies is limited [34]. HRV is a measure of cardiac autonomic function by counting the cyclic variations of RR intervals by an electrocardiogram, which reflects on autonomous nervous system functioning and serves as a measure to assess the body’s ability to modulate the physiological stress response. For example, an increase in normal-to-normal R-R interval, and normalized high frequency, which were associated with the reductions in depression, perceived stress, and anger, was found after a brief mindfulness and supportive group therapy for cancer patients [38].

There has been preliminary evidence that mindfulness may reduce aggression and violence, but none of these studies were based in the context of family caregiving [39]. In a recent trial, caregivers from the intervention group reported less but insignificant abusive behaviors toward care receipts [40]. Studies of MBI for family caregivers should include the outcome of conflict management, and the impact of interventions in preventing violence and maltreatment in the context of caregiving should be tested. Additionally, the spiritual well-being of caregivers on which people rely when encountering chronic or life-threatening illnesses are also important [41], and a study of MBSR for cancer patients reported a significant increase in spiritual well-being [42]. The proposed study will investigate the effects of MBI on several secondary outcomes, including HRV, family conflicts, and spiritual well-being.

Moreover, MBI studies should be expanded beyond symptom-based outcomes and establish evidence on indicators that are specific to MBI. One possible such indicator is experiential avoidance, which is defined as the unwillingness to remain in contact with distressing emotions, thoughts, memories, and physical sensations, at the potential expense of harm to the self [43]. Experiential avoidance is associated with a range of psychopathology, depressive and anxiety disorder. A study found no difference in experiential avoidance between dementia caregivers who received MBSR and a social support control group [23]. This may have been because the study was underpowered by a small sample size. A recent review of mainstream psychoeducation for caregivers reported that self-efficacy predicted positive outcomes [44], but the role of self-efficacy in MBI is uncertain. A trial of MBSR in Hong Kong mentioned earlier [25] found improvements in self-efficacy at post-intervention follow-up for mixed caregivers. The role of coping styles, experiential avoidance and self-efficacy in mediating the effects of MBI and psychoeducation will be tested in this study.

### 1.5. Research Questions and Hypotheses of the Study

This study will be a large clinical trial on Chinese caregivers of frail older adults comparing the effects of MBI and psychoeducation, with a treatment-as-usual control group. Based on the previous discussion on the literature review, the research objectives of the proposed study will be to: (1) investigate the effects of a brief MBI for caregivers of frail older adults on depression, burden, anxiety, spiritual well-being, family functioning, and family conflict; (2) compare the effects of MBI with an evidence-based caregiver program on above measures; and (3) test the factors (coping styles, self-efficacy, and experiential avoidance) that contribute to the positive improvements of both caregiver programs.

The following hypotheses will be tested: (1) Caregivers who participate in the MBI condition will report significantly greater reductions in depression, burden, and anxiety, increases in spiritual well-being and family functioning, and decreases in family conflicts, compared with the treatment-as-usual group. (2) Caregivers in the MBI condition will report similar outcomes in depression, burden, anxiety, spiritual well-being, family functioning, and family conflicts, compared with the psychoeducation condition. (3) Compared with caregivers after psychoeducation, those in the MBI condition will report more accommodation and less problem solving, denial, and self-blame in coping styles. (4) Changes in MBI will be mediated by experimental avoidance and self-efficacy, while changes in psychoeducation will be mediated by self-efficacy only.

## 2. Materials and Methods

### 2.1. Study Design

We will test the efficacy of the MBI and develop a model of mediators in the MBI for intergenerational caregiving, using a multi-site, three-arm, randomized controlled trial, which is more rigorous than a traditional two-arm design [45]. The effects of the MBI for caregivers will be tested by comparing the MBI (Arm 1), with psychoeducation (PSY, Arm 2) and treatment-as-usual (TAU, Arm 3). Assessments will be scheduled pre-intervention (T0), post-intervention (T1), and at 6-month follow-up (T2). Program effects will be tested using both pairwise between-subject comparisons (Arm 1 vs. Arm 2, Arm 2 vs. Arm 3, and Arm 1 vs. Arm 3) and within-subject comparisons (measures at T0, T1, and T2). The between-group analyses will be adjusted for baseline scores and for factors related to outcomes, including the age and gender of the caregivers, the severity of depression, anxiety, caregiver burden, and the level of frailty of older adults. Ethics approval was obtained before the commencement of the study.

#### 2.1.1. Refinement of Design from a Pilot Study

A pilot study was conducted from August to September 2019. Twenty-four caregivers participated in a pre-test, post-test comparison study. In the moderate to severe depression caregiver subgroup (CES-D > 11), the improvement in depressive symptoms was marginally significant (t[12] = 1.80, *p* = 0.10). Caregivers showed significant or marginally significant improvements in self-efficacy (t[23] = 4.30, *p* = < 0.001). When two participants with low burden scores were excluded, we found significant reductions in two subscales of experiential avoidance, namely procrastination (t[21] = 2.17, *p* < 0.05), distraction and suppression (t[21] = 2.28, *p* < 0.05). We had a follow-up meeting with social workers from a collaborating NGO, and feedback collected from participants and social workers were used to refine the MBI content.

#### 2.1.2. Sample Size Estimation

The sample size calculation is based on a previous study of an MBI for caregivers, in which an effect size of 0.64 in depression [25], with an estimation of an effect size of 0.14 for Arm 3. For a two-tailed α error of 5%, an 80% power, and a test of three independent groups, the required sample size will be 192 participants for three arms [46]. We further adjust the sample size based on an estimation of drop-out rate and intra-class correlation. An estimation of a drop-out rate of 15% is based on two local studies of mindfulness-based intervention [25,47]. Additionally, an estimation of intra-class coefficient of 0.07 is based on PI’s two recent mindfulness multi-site studies [48,49], and 240 caregivers will be recruited for this study. Based on such estimation, it should also be necessary to conduct mediational studies with 0.8 statistical power [50].

#### 2.1.3. Recruitment of Participants

The inclusion criteria for participation in this study are: (1) Caregivers of frail older adults based on a professional’s assessment of Clinical Frailty Scale with a score of 6, indicating a moderate level of frailty or above [51]. (2) Caregivers being adult children, or extended family members, e.g., children in-law of the elder care receiver. (3) Caregivers who are experiencing caregiver stress at the time of study with a score of 8 or above in Zarit Burden Interview Screening Version (ZBI-4) [52]. The exclusion criteria include: (1) caregivers who have diagnoses of psychosis, developmental disabilities, or cognitive impairment, which may present difficulties in comprehending the content of the program; (2) spouses, siblings, or friends will be excluded as their burden or depressive symptoms may be attributed to different reasons; (3) caregivers of older adults with moderate to severe dementia will be excluded, and the Clinical Dementia Rating Scale will be administered [53]; and (4) caregivers who have participated in an MBSR or equivalent will be excluded. Written consent will be collected for all eligible participants.

#### 2.1.4. Randomization

The research assistant, who will be blinded to the personal data of the participants, will administer the random assignment using computer-generated programming. Participants will be randomly assigned to MBI (Arm 1), PSY (Arm 2), or TAU (Arm 3). During recruitment and implementation, both Arms 1 and 2 are called “Family Psychoeducation Program”, and the term “mindfulness” will not be used for Arm 1 to minimize the potential placebo effect. A recruitment flowchart is attached in Figure 1.

### 2.2. Proceudres and Intervention

The research project will be announced and promoted among local collaborating NGOs and through advertisement in social media, promotional emails, and project leaflets. Currently, four NGOs have indicated their intentions to contribute to this study by assisting in promotion, recruitment, program implementation, and data collection. We will organize briefing sessions for explaining the rationale and procedures of the study twice yearly. All interested caregivers will be screened using Clinical Frailty Scale, Clinical Dementia Rating Scale, and caregiver stress.

The themes and content of Arms 1 and 2 are summarized in Table 1. Based on the protocols of PI’s previous clinical trials, brief mindfulness exercises will be integrated with psychoeducation of caregivers. The protocol for Arm 1 was developed in mid-September 2019 for the first pilot study.

The research team will modify the START (STrAtegies for RelaTives) program for caregivers of dementia [40] into a 10-h group program. Changes include the content which originally relates to the caregiver needs of dementia will be modified into frail diagnoses. Two hours will be added to allow for more discussion of individual caregiver’s concerns. The protocol for Arm 2 was developed in early 2021 and was endorsed by one of the authors who is an expert in caregiver intervention. Arms 1 and 2 are parallel in program design. Both are 4-session programs, administered bi-weekly and lasting for 10 h in total. Both programs include 15-min daily home practice assignments, with Arm 1 being guided with mindfulness exercises and Arm 2 with stress management knowledge. Both arms will be delivered in group format, with 10 to 15 caregivers in each group. Programs will be conducted in service units of four NGO collaborators and PI’s university. For caregivers in Arm 3, support is given as usual when their frail older adults receive community care service and regular consultation with general practitioners and specialists.

All instructors for Arm 1 require basic professional training in MBI plus at least two years of experience in conducting MBI. Instructors for Arm 2 will be recruited from NGOs with experience in working with caregivers of frail older adults for more than two years.

After the pre-intervention assessment (T0), caregivers who meet inclusion criteria will be randomized into a MBI (Arm 1), psychoeducation (PSY, Arm 2), or treatment-as-usual (TAU, Arm 3). After the intervention programs, participants in three arms will complete the post-intervention assessment (T1). A 6-month follow up (T2) is offered as a booster and final assessment for Arms 1 and 2. This study will be single-blinded, and outcome assessors will be blinded to randomization status. Cash remuneration coupons will be provided to caregivers who complete the study at T1 and T2.

To ensure intervention fidelity, all program sessions will be audio-recorded, and an independent rater will listen to 20% of randomly selected clips and assess whether each element in the intervention protocol has been implemented with consistency. Higher concordance rates will signify greater fidelity to the intervention protocol, which will be carefully monitored throughout the overall study. The treatment fidelity of Arm 1 will be further assessed by the Mindfulness-based Interventions-Teaching Assessment Criteria Scale [54]. Those of Arm 2 will be assessed by a checklist for implementing the START program.

### 2.3. Measures

In line with the study aims and hypotheses, the primary outcome variable is caregiver depression. Secondary outcome measures include caregiver burden, anxiety symptoms, spiritual well-being, perceived family functioning, family conflicts, conflicts between caregivers and care recipients, and heart rate variability. We will test for mediators of outcomes that include coping styles, self-efficacy, and experiential avoidance. All measures will be collected at three time points (T0, T1, and T2).

#### 2.3.1. Outcome Measures

The depressive symptoms of caregivers will be assessed using the 10-item version of the Center for Epidemiologic Studies Depression Scale (CES-D) [55]. Caregivers will be invited to indicate how they felt based on their experiences in the previous week using a 4-point scale from 0 (rarely or none of the time, less than 1 day) to 3 (all of the time, 5–7 days). Sample items include “My sleep was restless” and “I felt lonely.” A cutoff score of 10 is used for clinical depression. The CES-D was previously validated in a Chinese sample with a good internal consistency of 0.79 [56].

The Zarit Burden Interview is a 22-item measure of caregivers’ perceived stress level (ZBI) [57]. The degree of burden is measured across areas including health, psychological well-being, finances, social life, and relationship with the older adults. Sample items include “Do you feel that because of the time you spend with your relative, you don’t have time for yourself?” and “Do you feel that you have lost control of your life since your relative’s illness?” The caregivers will be asked to indicate the level of discomfort surrounding this question by choosing an answer ranging from 0 “not at all” to 4 “extremely.” The total score range is from 0 to 88. A score of 17 or more was considered high burden [52]. The scale has been validated among caregivers of patients with schizophrenia in Chinese with a high internal consistency (Cronbach’s alpha) of 0.88 [58].

The Hospital Anxiety and Depression Scale, Anxiety subscale is selected to measure the caregivers’ anxiety symptoms (HADS-A) [59]. Caregivers rate their symptoms from 0 “low” to 4 “severe,” and the anxiety symptoms score can range from 0 to 21. A score of 8 or above is considered clinical anxiety that deserves attention. HADS was validated and the internal consistencies for anxiety subscale were 0.77 [60].

The Functional Assessment of Chronic Illness Therapy–Spiritual Well-Being Scale (FACIT-Sp-12) [41] is selected to measure the spiritual wellbeing of the caregiver. It has 12 items, with two subscales in meaning/peace and faith and the sum of FACIT-Sp-12 scores from 0 to 48. The scale has been validated in a sample from Taiwan Chinese with a high internal consistency of 0.84 and 0.94 in two subscales respectively [61].

The 5-item Family Adaptation, Partnership, Growth, Affection, Resolve (APGAR) Scale was used to assess perceived family functioning [62]. Participants were asked to report their satisfaction with family functioning across five domains using a 3-point response scale, ranging from 0 = hardly ever to 2 = almost always. A sample item is: “I am satisfied that I can turn to my family for help when something is troubling me.” A Family APGAR score of less than 6 would be considered as a dysfunctional family. The scale has been validated in a sample from Hong Kong Chinese with a high internal consistency of 0.86 [63].

Family conflict was measured by the Conflict Tactics Scale (CTS) [64] a self-report measure of behavioral assault or psychological aggression among family members. The scales have eight items, including three subscales: verbal conflict (1 item), physical conflict (5 items), and positive negotiation (2 items). The Chinese version of the CTS2 scale was translated and validated in a previous study [65].

Heart rate variability is a measure of cardiac autonomic function in which the cyclic variations in the inter-beat intervals on an electrocardiogram are counted. It is also an early marker of cardiovascular risk [66]. It will be measured by the CorSense heart rate variability monitor.

#### 2.3.2. Measures for Mediating Variables

The Brief COPE scale is a self-reported measure of coping strategies [67]. The 28 items assess how frequently the participant engages in specific types of coping behavior or thoughts using a 4-point Likert scale ranging from 0 (I haven’t been doing this at all) to 3 (I’ve been doing this a lot). Sample items include “accepting the reality that the stressor happened and learning to live with it” and “refusal to believe the stressor exists.” The Chinese version of the Brief COPE was validated in a previous study with 11 subscales (problem-solving, accommodation, support-seeking, behavioral disengagement, denial, self-distraction, self-blame, humor, venting, substance use, and religion) [29]. It showed a satisfactory internal consistency of 0.76 for the whole scale, and those for subscales ranged from 0.49 to 0.87.

The 18-item Chinese version of the Caregiver Inventory (CGI-18) was used to assess caregiving self-efficacy in palliative care [68,69]. It consists of three domains: “care of the care recipient,” “managing information and self-care,” and “managing emotional interactions with the care recipient.” Each item is scored using a 9-point scale, from 1 (“not at all”) to 9 (“totally confident”). A sample item of CGI is “providing emotional support for the person I’m caring for”. The CGI has reported good internal consistency in Chinese samples (Cronbach’s alpha = 0.84 to 0.90).

Experiential avoidance will be measured by the 15-item Chinese Brief Experiential Avoidance (BEAQ) Questionnaire [70,71]. Each item is scored using a 6-point scale, from 1 (“strongly disagree”) to 6 (“strongly agree”). Sample items of BEAQ “When unpleasant memories come to me, I try to put them out of my mind” and “If I am in a slightly uncomfortable situation, I try to leave right away”. The BEAQ has reported a good internal consistency of 0.85 in Chinese samples.

#### 2.3.3. Program Fidelity

Mindfulness-Based Interventions–Teaching Assessment Criteria will be used to assess treatment fidelity of the mindfulness-based program. It includes six domains of competence in instructing a mindfulness program that might also apply to a brief mindfulness-based intervention [54]. All sessions will be audio recorded for assessing treatment fidelity. One of the four sessions (25% of the total program) will be randomly selected for each group. An independent reviewer who was an experienced teacher of mindfulness-based program with full professional training in MBCT will rate the implementation fidelity of the present study.

### 2.4. Data Analyses

All analyses will be carried out according to the intent-to-treat approach and all participants who receive randomization and allocation to one of three treatment arms will be included for analyses [72]. Two-way ANOVA will be used to evaluate the effects of the MBI (Arm 1), relative to PSY (Arm 2) and TAU (Arm 3), and the analyses of the primary and secondary outcome measures will be analyzed by comparing the values at T0 and T1. In addition to the immediate program effects, outcomes measured at T1 and T2 will be compared, to assess whether maintenance effects will be sustained at 6-months.

Mediation effects on the relationships among groups (MBI or PSY vs. TAU) variables and caregiver outcome variables (ZBI, CES-D, HAD-A) will be evaluated while controlling the baseline variables (sex and age of caregivers, frailty score) using PROCESS macro in SPSS [73], in order to compute the effects of the independent variable on the mediator (a), the effect of the mediator on the dependent variable (b), total effect (c), direct effect (c’), and bootstrapped (i.e., 10,000 random samples) bias-corrected 95% confidence intervals of the indirect effect (ab). According to Hayes [73], with significant paths a and b, even without a significant path c, confidence intervals that do not include zero indicate a significant indirect (i.e., mediation) effect.

## 3. Discussion

Family members play a pivotal role in the care of frail older adults in Chinese societies. They often experience a strong sense of obligation and burden for maintaining a quality care when the level of frailty of older adults is getting severe. Some caregiver programs have demonstrated their positive effects in reducing caregiver burden and depression, but limitations have been identified, such as long duration and intensive design that may not be feasible for implementation. There have been calls for a time-limited but more cost-effective program to mitigate the hardship under the realities of frail older adults and their families.

The proposed study is a multi-site, three-arm randomized controlled trial of a mindfulness-based intervention for Chinese family caregivers. Such rigorous study will generate evidence and knowledge that would have research and practice implications. Firstly, it will provide comparative effectiveness data on outcome variables for mindfulness-based interventions and psychoeducation on caregivers of frail older adults. We shall investigate the effects after intervention and at 6 months post-randomization using multiple outcome measures, including heart rate variability and family conflict that would offer advanced knowledge about the program effect on caregiver’s physiology and the risk of abuse. The low intensity of the intervention will provide a sustainable treatment option for policymakers, service providers, family caregivers, and other stakeholders.

Secondly, the investigation of mediators in this trial may also help to expand the existing models of intergenerational caregiving in Chinese culture, and to improve the overall quality of life of informal caregivers. The program effects on caregiver’s coping styles, self-efficacy, and experiential avoidance and their relationship with primary outcome will be tested. It would be helpful to study the effects of intergenerational caregiving who have been burdened by Confucian filial beliefs and compare the effects of two distinctive programs. Intervention theory in caregiving can be refined by comparing the outcomes of mindfulness-based intervention, which emphasizes caregiver stress reduction and self-care, with a psychoeducation program, which focuses on skills training and problem solving on their avoidance coping strategies.

## 4. Conclusions

Findings of this three-arm randomized controlled trial will contribute to the establishment of evidence base of mindfulness-based intervention in Chinese family caregivers. It may further shed light on the development of caregiver interventions in reducing caregiver burden and depression under the context of Confucian filial beliefs in Chinese populations and other similar societies that are influenced by collectivism.

## Figures and Tables

**Figure 1 ijerph-19-05447-f001:**
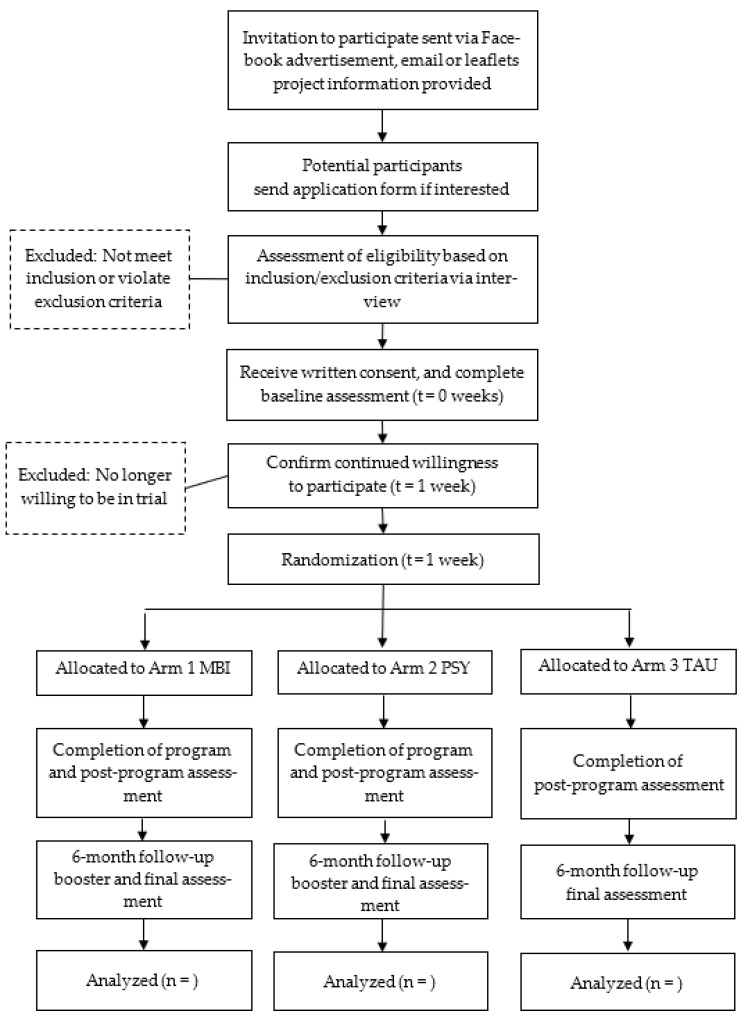
Recruitment flowchart.

**Table 1 ijerph-19-05447-t001:** Content of mindfulness-based intervention (Arm 1) and psychoeducation (Arm 2).

Mindfulness-Based Intervention (Arm 1)	Psychoeducation (Arm 2)
Session 1 Stepping out from automatic pilot Course orientation and mutual understanding Mindfulness exercise: Mindful eating and body scan	Session 1 Stress of caregivers Orientation and mutual understanding Overview of frailty Listing out problem Understanding Trigger-behavior-reaction chain Signal breath
Session 2 Reaction vs. responding to older adults Awareness of avoidance and aggression in caregiving Mindfulness exercise: Three-minute breathing and mindful stretching	Session 2 Making a behavior plan Setting behavior goals Changing behaviors by changing reactions Changing unhelpful thoughts Guided imagery
Session 3: Challenges in caregiving Awareness of pleasant and unpleasant events in caregiving Mindful communication Mindfulness exercise: Mindful sitting and ice cube exercise	Session 3 Communicating with older adults Communication styles (passive vs. aggressive vs. assertive) Assertiveness in communication Care options and pain management Relaxation exercise: Stretching
Session 4: Self-care and care of others Reviewing course learning Developing self-care plan Mindfulness exercise: befriending exercise	Session 4 Self-care and care of others Identifying pleasant events Course review

## Data Availability

Not applicable.

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
