# Peer review of "Mindfulness-Based Intervention for Caregivers of Frail Older Chinese Adults: A Study Protocol"

_ijerph, 2022, doi:10.3390/ijerph19095447_

Round 1
Reviewer 1 Report
Paper deals with a very relevant topic, with potential contribution to clinical intervention for frail older adults and their caregivers. some minor consideration could be given.
Introduction is well written and tried to summarized state of art. nevertheless, the amount of info given could be somehow confusing. Sincerely, until aim and hypothesis section I did not understand which variables were the principal outcome and which ones the secondaries. For similar reason, I suggest to enrich aim and hypothesis section in order to better explain it.
Method section. Given the complex design, I think that a better distinction between participant and procedure section could help the reader.
Moreover, Table 1 describes the content of MBT and PSY interevention. Could you better give also some info on TAU? sure, it represents the control group, but it could be useful know if basic info are given (they could differ from other centres)
Another concern regards the measures. I did not understand which measure were used and when. Are they administered in all step? or someones are only at baseline, or post-program?
Finally, please check all abbreviations. for example PSY, TAU, etc they could compare without a previous description.
Author Response
Paper deals with a very relevant topic, with potential contribution to clinical intervention for frail older adults and their caregivers. some minor consideration could be given.
Introduction is well written and tried to summarized state of art. nevertheless, the amount of info given could be somehow confusing. Sincerely, until aim and hypothesis section I did not understand which variables were the principal outcome and which ones the secondaries. For similar reason, I suggest to enrich aim and hypothesis section in order to better explain it.
Response: Thank you for your comment. We have reorganized the introduction, and the shorten the length. We also expanded section 1.5 research questions and hypotheses to improve the clarity.
Method section. Given the complex design, I think that a better distinction between participant and procedure section could help the reader.
Response: Thank you for your comment. We have moved two paragraphs from participants to procedures to improve the clarity.
Moreover, Table 1 describes the content of MBT and PSY intervention. Could you better give also some info on TAU? sure, it represents the control group, but it could be useful know if basic info are given (they could differ from other centres)
Response: Three arms are based on the same recruitment and randomization procedures, and they are from the same centres. More description is provided to arm 3: “For caregivers in Arm 3, support is given as usual when their frail older adults receive community care service and regular consultation with General Practitioners and Specialists.”
Another concern regards the measures. I did not understand which measure were used and when. Are they administered in all step? or someones are only at baseline, or post-program?
Response: Thank you for your comment. I have added one sentence to improve the clarity: “All measures will be collected at three time points (T0, T1, and T2).” in Section 2.3.
Finally, please check all abbreviations. for example PSY, TAU, etc they could compare without a previous description.
Response: Thank you for your comment. I have elaborated the use of abbreviation (PSY, and TAU) in the first paragraph of 2.1 study design.
Reviewer 2 Report
General note
Generally, caregivers of patients and frail older adults face a high level of burden that is usually underappreciated. When this burden is unattended, it can adversely affect rehabilitation outcomes and the well-being of the care recipients. Hence, interventions targeting the burden of caregivers are always a welcome development. This is even more apt with caregivers of older adults considering the current global socio-demographic shift towards old age. Generally, authors need to improve on their grammatical constructions. Below is section-by-section comments.
Title: Good title
Abstract: Well-written
Background:
Although this section highlighted the problems, it is rather too long. The authors were too verbose, and this would tire out the readers and make them lose interest. Authors should please summarise, shorten and better organize the background. This could be achieved by refraining from extensively discussing each study. Authors may also not need to have subheadings in this section.
Materials and Methods:
- It is nice to note that the authors already conducted a pilot study.
- The proposed sample size is adequate.
- Under the “Recruitment of participants”, in item 2, what are those other reasons burdens of spouses, friends, or siblings could be attributed to?
- What is the justification for exclusion criteria number 3
- Is this supposed to be a nationwide study? If not, the setting should be named and discussed.
- After figure 1, please provide a subheading “Interventions” before the write-ups on page 8.
- Page 8 (lines 324 and 325) stated that the protocol for psychoeducation would be developed and pilot-tested in early 2020. Has this been done? If yes, please provide the full protocol and the results of the pilot testing.
- Please authors should describe “treatment-as-336 usual” intervention. What does it contain?
- The statement on ethical approval (page 9, line 349) came rather too late. This should be stated higher up in the section of study design.
- 3 Measures (page 9, line 352): Table 1 does not contain the measures as stated by the authors.
- Line 359: “The depressive symptoms of caregivers WERE….” Has the study been performed? Authors had been using future tenses but resorted to past tense here.
- Scoring and interpretation of most outcome measures were not provided. The authors only reported the internal consistency of all the outcome measures. What validities and other psychometric properties?
- I seriously doubt if repeated measure ANOVA could comfortably analyze the data. Rather, a two-way ANOVA would be better.
Discussion: Well-written
Conclusion: Good
References: There are too many references. Authors can reduce this. Too many references came from a too-long background of the study.
Author Response
General note
Generally, caregivers of patients and frail older adults face a high level of burden that is usually underappreciated. When this burden is unattended, it can adversely affect rehabilitation outcomes and the well-being of the care recipients. Hence, interventions targeting the burden of caregivers are always a welcome development. This is even more apt with caregivers of older adults considering the current global socio-demographic shift towards old age. Generally, authors need to improve on their grammatical constructions. Below is section-by-section comments.
Title: Good title
Abstract: Well-written
Background:
Although this section highlighted the problems, it is rather too long. The authors were too verbose, and this would tire out the readers and make them lose interest. Authors should please summarise, shorten and better organize the background. This could be achieved by refraining from extensively discussing each study. Authors may also not need to have subheadings in this section.
Response: Thank you for your comment and your helpful suggestion to improve the background section. We have re-organized the introduction section and the word counts have reduced over 10%.
Materials and Methods:
It is nice to note that the authors already conducted a pilot study.
The proposed sample size is adequate.
Under the “Recruitment of participants”, in item 2, what are those other reasons burdens of spouses, friends, or siblings could be attributed to?
Response: Many intergenerational caregivers are influenced by filial piety and cultural values, while spouses, friends or peers, or siblings may be affected by deterioration in their own health conditions. Since the research team assume that caregiver burden of different caregivers may be different, we exclude caregivers in same generation in this study.
What is the justification for exclusion criteria number 3
Response: in order to test the outcomes of an intervention program, we exclude those caregivers who experience minimal level of burden as they are less likely to benefit from the intervention due to ceiling effect.
Is this supposed to be a nationwide study? If not, the setting should be named and discussed.
Response: This is a multi-site study but due to the unstable conditions of COVID, the exact sites will be reported in the final report and the next manuscript when it is completed.
After figure 1, please provide a subheading “Interventions” before the write-ups on page 8.
Response: we change the subheading of 2.2 into interventions as suggested.
Page 8 (lines 324 and 325) stated that the protocol for psychoeducation would be developed and pilot-tested in early 2020. Has this been done? If yes, please provide the full protocol and the results of the pilot testing.
Response: Thank you for your comment. The description of Arm 2 intervention has been revised as follow: “The protocol for Arm 2 was developed in early 2021 and was endorsed by one of the authors who is an expert in caregiver intervention.”
Please authors should describe “treatment-as-usual” intervention. What does it contain?
Response: The following description of treatment-as-usual is added: “For caregivers in Arm 3, support is given as usual when their frail older adults receive community care service and regular consultation with General Practitioners and Specialists.”
The statement on ethical approval (page 9, line 349) came rather too late. This should be stated higher up in the section of study design.
Response: As suggested, the ethical approval statement have been moved to the end of 2.1 study design.
3 Measures (page 9, line 352): Table 1 does not contain the measures as stated by the authors.
Response: Sorry for the confusion. The sentence has been removed.
Line 359: “The depressive symptoms of caregivers WERE….” Has the study been performed? Authors had been using future tenses but resorted to past tense here.
Scoring and interpretation of most outcome measures were not provided. The authors only reported the internal consistency of all the outcome measures. What validities and other psychometric properties?
Response: Thank you for pointing out the typo. “Were” has been changed into “will be”. Interpretation of scoring has been added to the measures of depression (CESD-10), caregiver burden (ZBI), anxiety (HADS-A), and family functioning (APGAR).
I seriously doubt if repeated measure ANOVA could comfortably analyze the data. Rather, a two-way ANOVA would be better.
Response: We have revised the method to two-way ANOVA in 2.4 data analysis.
Discussion: Well-written
Conclusion: Good
References: There are too many references. Authors can reduce this. Too many references came from a too-long background of the study.
Response: Thank you for your comment. We have reduce the references from 80 to 73.
Reviewer 3 Report
Study protocol is very reveal as caregivers of frail older adults are usually forgotten. Mindfulness is a rather new concept in other countries, so the planned study protocol will be very significant. Mindfulness could be implemented in health care.
Interventions in Table 1 presented very clearly. In line 352 written "all measures used are summarized in Table 1", but table 1 (line 331) is content of interventions NOT measures? Can authors specify?
As for international readers will be interesting to know about Arm 3 it is treatment -as-usual. Arm 1 and Arm 2 is explained very detailed, can authors more add about Arm 3 and what in their country mean treatment-as usual?
Author Response
Study protocol is very reveal as caregivers of frail older adults are usually forgotten. Mindfulness is a rather new concept in other countries, so the planned study protocol will be very significant. Mindfulness could be implemented in health care.
Response: Thank you for your recognition of the significance of the study.
Interventions in Table 1 presented very clearly. In line 352 written "all measures used are summarized in Table 1", but table 1 (line 331) is content of interventions NOT measures? Can authors specify?
Response: Sorry for the confusion. The sentence has been removed.
As for international readers will be interesting to know about Arm 3 it is treatment -as-usual. Arm 1 and Arm 2 is explained very detailed, can authors more add about Arm 3 and what in their country mean treatment-as usual?
Response: Thank you for your comment and suggestion. The following description is added in the revised manuscript: “For caregivers in Arm 3, support is given as usual when their frail older adults receive community care service and regular consultation with General Practitioners and Specialists.”